# Self-directed learning in health professions: A mixed-methods systematic review of the literature

Joana Berger-Estilita[1]*, Linda Krista[1], Artemisa Gogollari[1], Felix Schmitz[1], Achim Elfering[2,3], Sissel Guttormsen[1]

1 Institute for Medical Education, University of Bern, Bern, Switzerland, 2 Department of Work and Organizational Psychology, University of Bern, Bern, Switzerland, 3 National Centre of Competence in Research, Affective Sciences, University of Geneva, CISA, Geneva, Switzerland

* joanamberger@gmail.com

## Abstract

### Purpose

This study investigated the application and effectiveness of various self-directed learning (SDL) models in healthcare organizations. This study aims to identify the prevalent SDL models and factors influencing SDL adoption in healthcare settings.

### Methods

A systematic review was conducted, encompassing a comprehensive search across multiple academic databases (MEDLINE, Embase, PsycINFO, ERIC, and the Cochrane Library). The final search was conducted on April 16, 2024. The inclusion criteria were studies involving health and allied health professionals in clinical settings that explored SDL in any context or form of activity, emphasizing the description and/or use of an SDL model or SDL-related concept. Both qualitative and quantitative studies were included. We also explored the factors facilitating or hindering SDL and specific SDL-related outcomes, accommodating various study designs.

### Results

The final review synthesized findings from 34 articles involving over 5,700 healthcare professionals (including nurses, pharmacists, and physicians). The findings reveal that supportive organizational cultures significantly enhance SDL practices, whereas restrictive policies hinder their effectiveness. Various SDL models were identified and examined, showing that integrative frameworks combining individual motivation with structured organizational support yielded the best outcomes in fostering lifelong learning and adaptability among healthcare professionals. These findings emphasize the significance of individual motivation, learning environment, technological resources, social interactions, and SDL readiness among healthcare professionals.

provided the original author and source are credited.

**Data availability statement:** All relevant data are within the manuscript and its Supporting information files.

**Funding:** The author(s) received no specific funding for this work.

**Competing interests:** JBE and SG are associate editors for BMC Medical Education. JBE has received travel expenses from Medtronic for the Save the Brain Initiative training. This does not alter our adherence to PLOS ONE policies on sharing data and materials. The remaining authors declare that they have no known competing financial interests or personal relationships that could have appeared to influence the work reported in this paper. (as detailed online in our guide for authors http://journals.plos.org/plosone/s/competing-interests).

## Conclusion

Our findings demonstrate that SDL is crucial for the continuous professional development of healthcare providers and should be strategically supported by healthcare organizations. This study contributes to a better understanding of the interplay between SDL, workplace dynamics, and digital technology in healthcare practices. Identifying prevalent SDL models and factors influencing their adoption offers valuable insights for healthcare professionals on effective implementation strategies that address both its barriers and facilitators.

## Systematic review registration

reviewregistry1309, February 28, 2022 (www.researchregistry.com).

## Introduction

The demand for continuous professional development (CPD) is increasing in today's rapidly evolving healthcare landscape. Advances in medical science and technology have significantly shortened the half-life of medical knowledge [1], driving healthcare professionals (HCPs) to engage in lifelong learning to maintain their clinical competence and provide optimal patient care [2–4]. Self-directed learning (SDL) has emerged as a pivotal mechanism in this endeavor, enabling HCPs to keep pace with advancements and adapt to changing healthcare environments [5]. SDL is an instructional strategy in which learners decide what and how to learn. It can be completed individually or through group learning; however, the overall concept is that students take ownership of their learning [6].

While SDL is widely recognized at the individual level [7–10], its integration within organizational settings—particularly healthcare institutions—presents a complex interplay of challenges and opportunities that are less understood. Healthcare organizations vary greatly in how they support or impede the SDL efforts of their staff and are influenced by factors such as institutional culture, resource availability, and technological infrastructure [11–14]. These organizations play a critical role in shaping learning dynamics by either facilitating access to learning resources and opportunities or presenting barriers that can hinder the effectiveness of SDL practices.

Despite the acknowledged importance of SDL in medical education and practice, a significant gap remains in the understanding of how these organizational factors specifically influence SDL [15]. Most studies in this topic area focus on SDL as an individual endeavor, with less attention paid to how organizational contexts impact, shape, and potentially enhance it. This oversight is critical, as the success of SDL initiatives often hinges on the broader organizational environment in which they are embedded. To address this gap, our systematic review explored the following research questions:

RQ1: *How are different models of SDL applied within healthcare organizations?*

RQ2: *What are the key factors influencing the success of SDL in healthcare organizations, and how do these factors affect the experiences of healthcare professionals?*

By investigating these questions, we aimed to illuminate how healthcare organizations can better support SDL, enhancing the professional growth of healthcare providers, and how they can provide supporting structures for SDL. This review and analysis will help develop targeted strategies that healthcare institutions can implement to foster an environment conducive to SDL, ultimately contributing to improved healthcare outcomes.

## Methodology

### Study design

We systematically reviewed the literature on the SDL models applied by HCPs in healthcare organizations. This review followed the Preferred Reporting Items for Systematic Reviews and Meta-Analyses (PRISMA) recommendations [16] and the Johanna Briggs Institute (JBI) recommendations for mixed-methods systematic reviews [17]. Our study was designated as a mixed-methods review because of its comprehensive approach, integrating both quantitative and qualitative data from the included studies. Combining several methodologies allowed us to provide a more nuanced analysis of how SDL is implemented, perceived, and valued across different healthcare environments, enriching our understanding. The decision to adhere to the JBI recommendations for mixed-method systematic reviews, which combines quantitative and qualitative evidence, was driven by the need for a robust, transparent, and systematic framework that accommodates the complexity of combining diverse research types. This was essential for our review to successfully draw comprehensive conclusions regarding SDL practices across various healthcare contexts [17]. Grounded in Knowles's adult learning theory [6] and Bandura's self-efficacy theory [18], our empirical approach aligns with these frameworks by selecting outcome measures such as learner engagement and self-efficacy. The JBI methodology allowed us to synthesize rigorous evidence and account for variations in SDL interventions, ensuring robust and theory-driven conclusions. Our study protocol has been registered and published in the Review Register (no. reviewregistry1309; https://www.researchregistry.com/browse-the-registry#registryofsystematicreviewsmeta-analyses/).

### Inclusion criteria

*Studies:* The considered studies included health and allied health professionals in clinical settings. Health professionals include physicians, nurses, pharmacists, and other practicing HCPs who treat patients or provide direct patient care. A clinical setting refers to a location where the primary purpose is the delivery of patient care.

*Phenomenon of interest:* We selected studies that examined SDL in any form of activity. All studies that used the terms and definitions described above were included.

*Outcomes:* The primary outcome of inclusion was the description and/or use of a model for SDL and SDL (or a related concept) in general. Secondary outcomes were factors promoting or hampering SDL in healthcare organizations and specific outcomes related to SDL.

*Types of studies:* The following study types were considered: randomized controlled trials (RCTS), cohort studies, case-control studies, cross-sectional surveys, qualitative studies, mixed-methods studies, systematic reviews, narrative reviews, editorials, and letters to the editor.

### Exclusion criteria

*Irrelevant or incorrect topics:* Studies outside the scope of SDL or those unrelated to HCPs were excluded. Studies focusing on formal education and traditional learning processes were excluded.

*Incorrect Target Group:* Studies not involve HCPs in clinical settings and those involving students or residents in training were excluded, because these groups are heterogeneous and mostly have structured learning pathways.

*Inapplicable formats:* Studies presented as conference abstracts, report abstracts, congress papers, comments/replies, proceedings, or reference materials were excluded.

*Inappropriate journals:* Studies published in predatory or inappropriate journals were excluded. Beall's List of potentially predatory journals and publishers was used to identify and exclude studies published in predatory journals.

The inclusion and exclusion criteria were carefully applied to ensure that only relevant, high-quality studies were considered for the systematic review.

All methods of analysis and inclusion criteria were specified and documented in the protocol (see S1 Appendix).

## Search strategy

The final search was performed on the April 16, 2024 in Ovid MEDLINE and Ovid Embase, PsycINFO, ERIC, and the Cochrane Library (S2 Appendix). We first performed a general literature review on the topic of SDL and related terms [19] and used known SDL work psychology models to build the study's query (S3 Appendix). No restrictions were imposed on the search.

## Selection process

Four researchers (LK, AG, JBE, and FS) conducted the study selection independently, using EndNote 20 (EndNote Team, 2013), and all disagreements were resolved by consensus or through consultation with another investigator (SG) in the review team. Duplicate and irrelevant studies were rejected after examining the titles and abstracts. The first stage of the screening process was piloted using a sample of 100 articles. Based on the pilot screening, the criteria was refined and four more rounds with 100 articles were independently screened by two reviewers and categorized into EndNote as either a) included or b) excluded. Cohen's κ was run at every round to determine if there was agreement between the two reviewers on whether to include or exclude a study. Subsequently, the full texts of potentially eligible studies were obtained and reviewed according to the inclusion and exclusion criteria. Where the full text was initially unavailable, we contacted the authors directly to obtain the original articles. We obtained access to all the original articles for the selected studies. The PRISMA flow diagram (Fig 1) outlines the study selection process.

## Data collection

LK and SG developed the data extraction sheet based on the JBI mixed-methods systematic review guidelines [20]. Each study was appraised for inclusion based on predefined criteria that aligned with the JBI standards. Data from each report were extracted by two pairs of reviewers who worked independently. After independent data extraction, the reviewers compared and resolved any discrepancies in their data entry. In cases in which discrepancies could not be resolved through discussion, a third reviewer from the same group was consulted to make the final determination [20]. The template was pilot-tested on ten randomly selected studies and refined accordingly. Finally, the selected articles were sorted by study type before the data were extracted and entered into a spreadsheet. The following information, if available, was extracted from the included studies: General data (record number, reference title, author, year of publication, country of publication, type of study, and specific study classification), the phenomenon of interest, method (design, operationalization and measuring instrument), population studied (country of origin, professional background, and setting), theoretical background of the study, description of SDL (term used and definition), SDL models, factors affecting SDL, evidence-based outcomes of SDL, and other information (additional findings, conclusions, references to other relevant studies, and notes).

## Data synthesis

Following the article extraction, data were analyzed using SPSS 23.0. (IBM Corp., Armonk NY, USA). We reported descriptive statistics for the quantitative data (median, IQR). Our qualitative analysis explored the theoretical frameworks guiding the research, diverse conceptualizations of SDL, and contextual factors influencing its dynamics. Qualitative

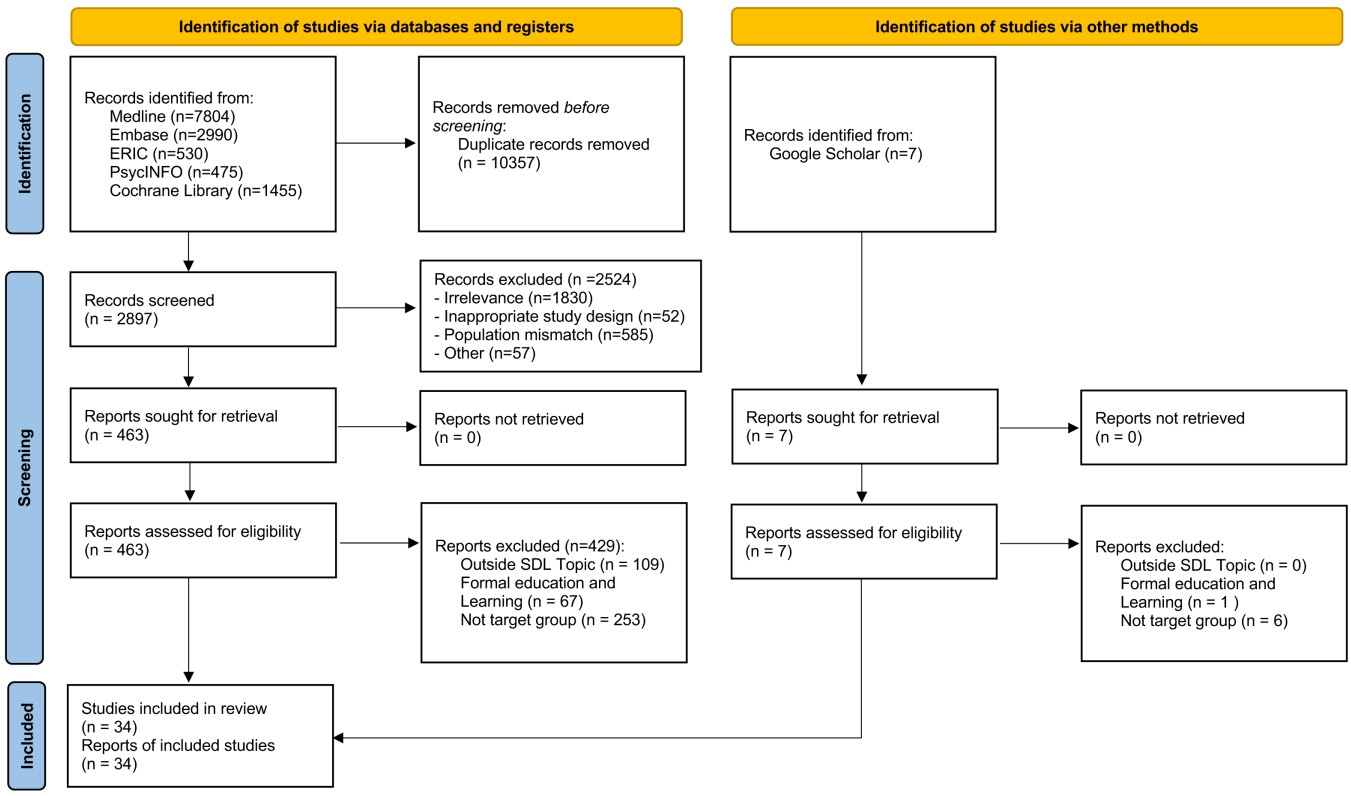

**Fig 1. PRISMA Flow diagram with search results.** Identification: Number of records identified through database searches and additional sources. Screening: Records screened after duplicates were removed. Eligibility: Full-text articles assessed for eligibility. Included: Studies included in the qualitative and/or quantitative synthesis (meta-analysis). Excluded: Number of records excluded at each stage and reasons for exclusion.

data from the selected studies were processed according to the Miles and Huberman framework for data analysis [21]: data segmentation, editing, and summarization, followed by data display. Themes related to the multifaceted determinants affecting SDL, including personal, organizational, and environmental factors, were identified through comprehensive content analysis. Data extracted were synthesized narratively, using an integrative and aggregative approach. The JBI recommendations for mixed-method systematic reviews guided our overall methodological approach, particularly in structuring the review, assessing the quality of the included studies, and synthesizing the findings. After applying the Miles and Huberman framework [21], we synthesized these qualitative results with the quantitative data following the JBI's convergent integrated approach to ensure that our findings holistically reflected both data types. The data synthesis was meta-aggregated using JBI SUMARI. After completing the appraisal process, each study's findings were grouped into themes, with an overall synthesized flowchart as a representative conclusion, following JBI recommendations. Each of the synthesized findings was designated based on the confidence level in terms of credibility and dependability.

## Statistical analysis

We performed a meta-analysis to examine the specific outcomes of the studies involving quantitative SDL outcomes. The meta-analysis was planned using RevMan software v.5.4.20, applying a random-effects model to account for expected heterogeneity across the included studies. Data should include means and standard deviations (SD), with the effect size estimated by the mean difference (MD) and reported with 95% confidence intervals (CI). Studies that did not report group

means or MD were excluded. Where meta-analysis could be conducted, we summarized and described the data in the text.

## Results

### Study selection

The literature search retrieved 13254 articles. After applying the inclusion and exclusion criteria and removing duplicates, 34 articles were included in the review (PRISMA Flow diagram, Fig 1) [22–56].

### Study characteristics

The basic characteristics of these articles are presented in S4 Appendix. Approximately 5723 participants across all studies were included, of which 54.0% (n = 3091) were nurses, 17.8% (n = 1023) were pharmacists, and 22.4% (n = 1273) were physicians. The median number of HCPs in the SDL intervention group was 216 [CI 50.5–330].

There were 11 (32%) qualitative, 18 (53%) quantitative, and 2 (6%) mixed-methods studies. The remaining studies were two reviews and one editorial.

Eight (24%) studies originated from North America (USA: 5; Canada: 3), 12 (35%) from Europe (Denmark: 1; Romania: 2; Sweden: 2; UK: 2; The Netherlands: 1; Norway: 1; Spain: 1; Belgium: 1; Turkey: 1) and 12 (35%) from Asia (Brunei: 1; Iran: 2; Japan: 1; China: 4, South Korea: 3; Taiwan: 1). The remaining two (6%) were from Kenya and Paraguay. Studies focused on various aspects of SDL, including factors influencing it, the impact of interventions, the role of the learning environment, assessment of SDL readiness, barriers and strategies for SDL, informal learning, lifelong learning, and the use of technology.

Data collection methods included surveys and questionnaires (n = 18, 53%), interviews (n = 8, 24%), observations (n = 2, 6%), and focus groups (n = 2, 6%). In the remaining studies, no method was present.

### Synthesis results

**Quantitative data.** There were 11 (32%) qualitative, 18 quantitative (53%) and 2 (6%) mixed-methods studies. The remaining studies included two reviews and one editorial.

Studies focused on various aspects of SDL, including influencing factors, the impact of interventions, the role of the learning environment, assessment of SDL readiness, barriers and strategies for SDL, informal learning, lifelong learning, and the use of technology.

Data collection methods included surveys and questionnaires (n = 18, 53%), interviews (n = 8, 24%), observations (n = 2, 6%), and focus groups (n = 2, 6%). Surveys are broader research tools designed to gather data from a sample population, typically through a set of questions aimed at eliciting information about respondents' characteristics, behaviors, or opinions. Surveys can be administered in various formats, including questionnaires.

Questionnaires are defined more specifically as surveys that consist of a series of questions written for the primary purpose of collecting responses. They can be used as stand-alone data collection tools or as part of larger surveys.

In this review, we opted for a narrative synthesis rather than a meta-analysis because of the significant heterogeneity observed across the included studies. The variability in study design, population characteristics, intervention types, and outcome measures made it inadequate to aggregate the data, as this would risk introducing bias. Studies have ranged from qualitative explorations to randomized controlled trials, each employing different methodologies and measuring diverse SDL-related outcomes in healthcare settings. Moreover, the heterogeneity in sample sizes, demographic profiles, and contexts presents a challenge for statistical synthesis. To preserve the integrity of the individual findings and avoid obscuring the complexity and nuances within the data, we synthesized the results narratively. This approach allowed us to

explore trends and gaps in the literature, highlight key themes, and provide a more detailed understanding of SDL models and factors in various healthcare environments, ensuring that the conclusions were contextually relevant and reflective of the diverse research landscape. Overall, combining studies with diverse methodologies risks producing misleading results and obscure meaningful differences in their effects [57].

**Qualitative data thematic analysis.** Our analysis and meta-aggregation identified eight main themes: (1) SDL models, (2) impact of interventions on SDL, (3) importance of social interactions and learning environment, (4) association between SDL and academic motivation/clinical performance, (5) informal learning and lifelong learning, (6) use of technology for SDL, (7) factors promoting SDL in healthcare organizations, and (8) barriers to SDL in healthcare organizations. The complete meta-aggregation results are reported in S5 Appendix (qualitative results) and in S6 Appendix (flowchart).

**Theme 1: SDL models.** Of the 34 selected studies, 12 (35%) explicitly used or referenced models or frameworks to guide their investigations of SDL and related phenomena. These provide the theoretical underpinnings and conceptual frameworks for understanding various aspects of SDL, including its facilitators, barriers, influencing factors, and outcomes. Researchers have applied these models to guide their investigations and analyze their findings in the context of SDL. Thirteen models were applied across the 12 studies, with three studies [22] using two models. The 12 studies can be classified into the following types: four combined models, five process models, three context models, and one personal model. The models and frameworks used in these studies are summarized in Table 1.

**Theme 2: Impact of interventions on SDL.** Several studies highlighted the positive effects of interventions on SDL [26,42,48,53]. Sockalingam [53] found a positive correlation between motivation and lifelong learning scores. Kim [48] demonstrated the impact of chatbots on satisfaction with learning content, usability, and intrinsic learning motivations among new nurses. Bing-Jonsson [42] also found a significant improvement in competencies after introducing a workplace-based blended learning program during the outbreak of the COVID-19 pandemic. Fahlman [26] highlighted the positive perceptions among registered nurses about using mobile devices for informal learning, showcasing their potential to support various aspects of professional development.

Collectively, these studies provide insights into the varying effects of interventions on SDL behaviors, motivation, goal-setting abilities, and time allocation among HCPs.

**Theme 3: Importance of social interactions and learning environment.** Several studies highlighted the significance of social interactions and learning environment in influencing SDL [24,30,47,52,55,56]. Gathu [47] explored how the learning environment influences graduate students' reflective practices. Papanagnou [52] examined the complexity of the clinical environment during the COVID-19 pandemic and its impact on team learning. Both Yao [55] and Wang [56] investigated the role of social interaction in the learning process of hospital pharmacists. Chakkaravarthy [24] indicated that workplace characteristics, such as experience, education, and task variety, influenced nurse readiness for SDL. In a dissertation, Lee [30] underscored the importance of individual and organizational factors, including workplace friendship, learning motivation, and organizational support, in shaping informal learning behaviors among nurses. These findings collectively emphasize the crucial role of the learning environment and social interactions in shaping SDL behaviors and attitudes among HCPs.

**Theme 4: Association between SDL and clinical performance.** Some studies have also explored the relationships between SDL and other constructs in the context of HCPs [27,31,47,52,53,55,56]. Sockalingam [53] demonstrated a positive correlation between motivation and SDL readiness among psychiatric HCPs, implying that higher motivation is linked to a greater inclination towards lifelong learning. Lim [31] suggested an indirect connection between faculty members' readiness to foster SDL and medical students' clinical performance, highlighting the potential impact of promoting SDL behaviors on students' practical skills. Gathu [47] suggested that a flexible and supportive learning environment enhances reflective practices, which are crucial for SDL and potentially improves clinical performance. Papanagnou [52] implied that understanding and adapting to clinical environments via SDL could lead to more effective

**Table 1. Description of models used in selected studies.**

| Name of Model | Type of Model | Description of Model | Context used |
|---|---|---|---|
| Deci and Ryan's Self-Determination Theory [58] | Personal Model | It describes intrinsic motivation and factors influencing learning behavior. | Sockalingam [53] investigated how participation in the ECHO-ONMH virtual CPD program could enhance learning beliefs and motivation for lifelong learning. |
| Cuyvers' Self-Regulation of Professional Learning model [59] | Process Model | Analyses self-regulated learning in the clinical environment, focusing on strategies such as questioning, alertness, recognizing learning needs and opportunities, and self-regulatory readiness. | Cuyvers [46] uses this model to investigate self-regulated workplace learning (SRwpL) strategies used by nurses in clinical wards. |
| Dreyfus and Dreyfus Model of Skill Acquisition [60] | Process Model | It describes the stages of skill acquisition from novice to expert. | Applied by Andersen [22] to understand how GPs learn POCUS, this model centers on the evolving process of skill development. |
| Knowles' model on self-directed learning [6] | Process Model | Defines SDL as a self-initiated process that may or may not require external assistance, involving steps like identifying learning needs, setting goals, finding resources, implementing strategies, and evaluating outcomes | In Liu [50] Knowles' principles are applied to define and measure nurses' ability to engage in self-directed learning, which includes identifying learning needs, setting learning goals, finding resources, applying learning strategies, and evaluating learning outcomes. |
| Slotnick's four-stage theory of professional learning [61] | Process Model | Describes a process where professionals first recognize a problem, then seek information, undergo a conceptual change in understanding, and finally apply behavioral changes to improve their work practices. | Hill [28] used this theory to frame the stages professionals go through when encountering and resolving complex challenges in their practice, providing a structured lens to examine their learning processes and adaptations in a dynamic, integrated care setting |
| Ten Cate's learning-oriented teaching (LOT) model [62] | Process Model | Emphasizes integrating cognitive, affective, and metacognitive strategies into teaching to promote self-regulated learning and prepare students for lifelong learning. | Used by Lim [31] to explore the preparedness of faculty in medical schools to support SDL among students. |
| Garrick's Model of Informal Learning [63] | Context Model | It emphasizes informal learning within organizations and the role of participation opportunities. | Referenced by Claret [25] for workplace learning strategies, it focuses on learning in the context of work environments. |
| Karasek's Job Demands-Control (JDC) model [64] | Context Model | Describes how the relationship between job demands, job control, and the level of social support affect an employee's health and wellbeing. | Lin [49] explored the effects of job characteristics on physicians' orientation toward lifelong learning, using this model to analyze how job demands and job control, along with social support from colleagues and supervisors, influence lifelong learning among physicians. |
| Marsick and Watkins Models for Informal and Incidental Learning [65] | Context Model | They focus on dimensions of informal and incidental learning in the workplace. | Lee [30] and Fahlman [26] applied these models to explore factors influencing informal learning in nurses, emphasizing the workplace environment's role in learning. In Papanagnou [52], the model was employed to analyze how clinical teams learn adaptively and informally in response to the dynamic and uncertain conditions of the COVID-19 pandemic. |
| Zimmerman's Self-Directed Learning Theoretical Model [66] | Combined Personal and Process Model | This model views self-directed learning as a dynamic and cyclical process where individuals actively engage in identifying their learning needs, setting goals, planning and implementing learning strategies, monitoring and regulating their learning process, and evaluating their learning outcomes. | Liu [50] used the model to develop a theoretical framework for the Nurses' Self-Directed Learning Competence Scale. |
| Bronfenbrenner's ecological systems theory [67] | Combined Personal and Process Model (broader environmental model) | Examines how individual growth is influenced by interacting layers of environmental contexts, ranging from immediate settings like family and school to broader societal and cultural forces (Micro-, Meso-, Exo-, Macro- and Chronosystem) | Clouder [45] uses it for its analytical potential to understand practitioners' experiences within nested contexts, recognizing how individual, organizational, and policy-level factors interact to influence learning outcomes in integrated care settings |

*(Continued)*

**Table 1.** (Continued)

| Name of Model | Type of Model | Description of Model | Context used |
|---|---|---|---|
| Lam Framework [68] | Combined Personal and Context Model | It takes into account individual and organizational factors in the learning process. | Andersen [22] used this framework to study how GPs learn POCUS, highlighting its emphasis on the organizational context of learning. |
| Tynjälä's 3P Model [69] | Combined Context, Personal and Process Model | Analyzes learning through three interconnected components—Presage (input factors like student characteristics and teaching context), Process (learning activities and strategies), and Product (outcomes of the learning process) | Used by Berg-Jansson [41] to explore how individual traits like commitment and situational factors such as managerial support impact workplace learning processes and outcomes, such as personal development and productivity, among temporary agency nurses |

CPD, continuing professional development; ECHO-ONMH, extension for community healthcare outcomes–Ontario mental health; GP, general practitioner; JDC, job demands control; LOT, learning-oriented teaching; POCUS, point-of-care ultrasound; SDL, self-directed learning; SRwpL, self-regulated workplace learning.

clinical performance. Yao [55] underscored the importance of practical application in reinforcing learning, indicating that SDL can improve hospital pharmacists' knowledge and clinical skills. Ghiyasvandian [27] and Yao [55] emphasized the critical role of applying theoretical knowledge to practical settings as part of SDL, highlighting that real-world applications are key to reinforcing learning and professional development. Collectively, these findings suggest that greater academic motivation could enhance engagement in SDL, possibly contributing to improved clinical performance and overall professional development among HCPs.

**Theme 5: Informal learning and lifelong learning.** Some of the studies delved into informal learning and its implications for SDL among HCPs [26,29,30,52]. Lee [30] found positive correlation between nurses' engagement in informal learning activities and higher clinical performance scores, emphasizing the potential of informal learning experiences to enhance practical skills. Kyndt et al. [29] revealed that opportunities for feedback and self-efficacy were pivotal in predicting successful learning outcomes from informal learning activities, highlighting the importance of conducive conditions and individual attributes for effective informal learning in the healthcare sector. Yao [55] illustrated informal learning strategies, such as summarizing, reflection, and integrating theory with practice. Andersen [22] emphasized the self-navigated learning process and informal experience gathering, while Berg Jansson [41] described the informal learning-related work and learning conditions of temporary agency nurses as they adapt to various client organizations and engage in self-directed professional development outside of structured programs.

**Theme 6: Use of technology for SDL.** Three studies investigated the impact of technology on SDL among HCPs [22,39,55]. Andersen's [22] study on general practitioners (GPs) demonstrated that technology, such as point-of-care ultrasound, played a crucial role in GPs' continuous learning process. Allen [39] highlighted how physicians use digital resources, such as the online database UpToDate, for continuing professional development and emphasized the role of technology in facilitating informal learning beyond formal education settings. Yao [55] provides a clear example of technology in SDL, showing how hospital pharmacists leverage online courses, professional databases, and digital platforms, such as forums and apps, to enhance their learning and professional development. These studies highlight the potential of technology, including tablets and videos, to enhance SDL experiences by enabling self-regulation, overcoming barriers, and fostering continuous learning among HCPs.

**Theme 7: Factors promoting SDL within healthcare.** Various factors have been identified as influential in shaping HCPs' SDL readiness [24,27,29–32,35], as shown in Table 2.

Ghiyasvandian [27] and Chakkaravarthy [24] highlighted nurses' overall high levels of readiness for SDL without significant correlations with age, gender, academic status, or marital status. Taylor [35] suggested a correlation between a lifelong learning orientation and the pursuit of knowledge among psychologists, indicating a connection between positive

**Table 2. Promoting factors and Barriers to SDL in healthcare organizations.**

| Factors | Promoting Factors | Barriers |
|---|---|---|
| **Personal Factors** | • Intrinsic motivation<br>• Confidence in SDL abilities<br>• Positive attitudes toward lifelong learning<br>• Self-efficacy<br>• Faculty engagement<br>• Friendship and workplace interactions<br>• Task variety and significance | • Lack of dedicated time for SDL activities<br>• Difficulties in self-assessment<br>• Reliance on clinical experiences<br>• Individual learner barriers and personal well-being circumstances<br>• Challenges related to the process of setting and pursuing goals<br>• Information overload and uncertainty about effective information |
| **Process Factors** | • Goal achievement strategies<br>• Support by digital tools<br>• Memorization, integration of theory and practice, reflection | • Excessive goal-setting or inadequacies in plan development<br>• Challenges in using digital and mobile technologies for SDL (e.g., information overload, uncertainty) |
| **Contextual Factors** | • Social interactions with peers, consultants, nurses, and patients<br>• Supportive organizational policies<br>• Access to digital tools and online resources<br>• Organized learning support and feedback | • Competition with peers<br>• Situational barriers, such as time pressure and inconsistent schedules<br>• Technical issues (e.g., digital connectivity, device specifications) |

attitudes towards SDL and the commitment to continuous learning. Malekian [32] found that intrinsic motivation and SDL readiness scores significantly impacted nurses' SDL readiness, emphasizing the role of internal motivation. Kyndt et al. [29] demonstrated that opportunities for feedback and higher self-efficacy were associated with greater success in both generic and job-specific learning outcomes among nurses, and that supportive organizational conditions and personal traits were key in predicting successful outcomes from informal learning activities. Personal factors, such as task variety, task significance, workplace friendship, learning motivation, and organizational factors, also influenced informal learning among nurses [30]. Both individual characteristics and the work environment played significant roles in shaping nurses' engagement in informal learning activities [29]. Lim [31] underscored the importance of faculty engagement in promoting SDL readiness among medical students, with faculty members using metacognitive and affective approaches to create an environment conducive to SDL.

Collectively, these findings suggest an interaction between intrinsic motivation, faculty involvement, and positive dispositions in fostering SDL readiness among HCPs. We identified three factors—Person, Process, and Context— as a structured lens through which to understand how individual traits, learning methods, and environmental influences interact to facilitate SDL readiness among HCPs.

*Personal factors*: Several studies delved into the personal factors promoting SDL among HCPs [23,27,30,32,55,56,70,71]. For instance, Kyndt et al. [29] identified self-efficacy as a pivotal predictor of both generic and job-specific learning outcomes, highlighting its importance in various organizational and functional contexts. Li et al. [7,8] found that factors such as confidence in SDL abilities, positive attitudes, and the propensity for lifelong learning were crucial for effective SDL. Some studies [8,23,70] noted that individual skills in self-regulatory mechanisms, such as emotion control, metacognition, attention focusing, and effort control, significantly promoted SDL. Sockalingam [53] underlined the importance of developing intrinsic motivation to foster lifelong learning behaviors. Wang [56] identified intrinsic motivations for SDL such as curiosity, interest in achievement, self-efficacy, and personal growth. Friendship, experience, and demography also emerged as factors influencing SDL. Lee [30] noted that collegiality with other nurses, learning motivation, and aspects, such as task variety and meaning, significantly affect informal learning among nurses. While several studies examine demographic variables, such as age, gender, and community setting [27,32,72], they did not show significant effects on SDL activities, highlighting that personal traits and attitudes may be more influential in enhancing SDL than demographic factors among HCPs.

*Process factors*: In the context of SDL, process factors refer to specific elements that contribute to the learning process and ultimately impact SDL outcomes. In the noted studies, two overarching factors emerged that play a role in supporting the SDL process:

1. Goal Achievement. This factor involves individual factors that influence SDL outcomes when individuals explicitly define their goals. Intrinsic motivations such as curiosity, interest in achievement, and self-efficacy directly influence the SDL process by encouraging individuals to define and achieve their learning goals explicitly [56]. Meanwhile, aspects such as memorization, integration of theory and practice, and reflection can be linked to goal achievement in SDL as they involve setting specific learning objectives and using personal cognitive approaches to meet them [55].

2. Support from digital tools. Digital tools are closely linked to SDL, which is crucial for facilitating the learning process. While most studies acknowledge their importance of digital tools, they often provide vague descriptions of their influence. Easy access to mobile devices is relevant for SDL activities, and the combination of technology use and online resources has been identified as a significant facilitator for SDL, offering easy access to up-to-date information and providing resources and platforms for information exchange and skill enhancement [26,39,55].

*Contextual factors*: Several studies have explored the contextual factors that promote SDL [27,30,32,47,52,56,73], revealing four overarching factors that influence SDL among HCPs, detailed below.

1. Social interaction. Peer and team interactions significantly shape HCPs' SDL activities. Peer influence and patient interactions motivate SDL among hospital pharmacists [56]. Daily work interactions catalyze SDL, emphasizing the role of peers and a broader team environment in the learning process [56]. Clinical teams also seem to adapt and learn informally in response to complex and chaotic situations [52]. Team dynamics and peer interactions in navigating these challenging environments are integral to SDL because teams must actively to adapt and manage their learning processes.

2. Learning environment. The learning environment has multiple dimensions that affect SDL. These include curriculum facilities, atmosphere, patient-related factors, available time, team engagement, and program support influence SDL [47,52,56]. Organized learning support and opportunities for feedback positively impact informal workplace learning [29]. Lee [30] emphasized the role of organizational care and communication in influencing informal learning.

3. Learning programs. The structure and support of learning programs impact SDL. Providing protected time for it, guidance from learning coaches, and problem-based learning curricula have shown positive effects on developing SDL skills [29]. However, the influence of learning program activities can vary across studies due to methodological differences and diverse understandings of the influencing factors [27,39,55].

4. Organizational support and digital tools. Supportive organizational policies and access to digital learning tools enhance SDL. Yao [55], Papanagnou [52] and Wang [56] explored how social interactions, conducive learning environments, and managerial support, including the use of digital tools, facilitate SDL among HCPs. These factors range from peer influence and professional forums to supportive organizational structures and access to professional databases and online courses. These elements create a robust framework that encourages continuous professional development and effective learning adaptations in complex healthcare environments. Other studies, such as those of Gathu [47] and Fahlman [26] offer insights into reflective learning and technological support but only encompass part of the full range of contextual factors affecting SDL.

**Theme 8: Barriers to SDL in healthcare organizations.** Several studies have examined barriers to SDL and the strategies employed by learners to overcome these challenges [24,25,29,30,41–43,46,48–50,52,53], as shown in Table 2. Identifying strengths and weaknesses, using methods to track progress, and having a propensity for lifelong learning were strategies associated with successful SDL. Barriers included lack of time devoted to SDL activities, difficulties in

self-assessment, and reliance on clinical/curricular experiences for learning competence. Similar to the factors that promote SDL, hindrances can be categorized into personal, processed, and contextual aspects.

***Personal factors***: Studies addressing personal barriers [27,32,54,56] revealed that individual learner barriers and personal well-being circumstances can act as impediments. Age, gender, academics, marital status, psychological resilience, and mindful agency are potential barriers [27,32,54]. Wang [56] focused on the intrinsic and extrinsic motivational factors influencing SDL among hospital pharmacists and identified personal motivations and interpersonal/work-related factors as possible barriers.

***Process factors***: Several studies highlighted challenges related to setting and pursuing goals [22,41–43,45,46,52,56], indicating that excessive goal-setting or inadequacies in plan development can hinder SDL. Other obstacles included competition among peers, low motivation from supervisors and colleagues, time pressures, inconsistent schedules, competing priorities in the clinical environment, and technical issues with digital tools and mobile devices. These obstacles hindered the effective implementation of SDL among HCPs and posed challenges to their ongoing professional development and learning.

***Contextual factors***: Contextual barriers are external factors that influence SDL but are outside the learner's control [41,45–47,52,55]. Obstacles include a lack of organizational sustainability and funding issues that affect the learning process, rigid instructional approaches that do not allow flexibility or creativity in learning methods, and challenges in adapting quickly to new and rapidly changing protocols during the COVID-19 pandemic. Technical issues, including digital connectivity and mobile device specifications were identified as barriers to using digital and mobile technologies for SDL in healthcare workplaces. The complex demands of the clinical environment, including patient care and research, create competing priorities that hinder SDL.

## Discussion

This systematic review of SDL across healthcare settings analyzed 34 studies involving 5723 participants, revealing significant insights into the factors influencing SDL among HCPs. The key findings highlight the critical role of organizational culture in facilitating or hindering SDL, the effectiveness of specific interventions in enhancing SDL competencies, and the profound impact of learning environment on SDL readiness.

Our study and insights related to the first RQ (*How are different models of SDL applied within healthcare organizations?*), have implications on several levels. First, on the *theoretical level* the results demonstrated a significant improvement in self-efficacy scores, supporting Knowles' theory of adult learning [6], which posits that autonomy leads to enhanced learner confidence. Interestingly, engagement scores showed only marginal improvement, suggesting that other factors, perhaps not covered by the theory, may play a role in engagement during SDL interventions. These findings highlight the need to further refine the theoretical model to account for additional motivational factors. A further theoretical implication is that this review also highlighted gaps in the application of SDL models, with only about one-third of the included studies exploring implementation in clinical settings. While some studies referenced frameworks such as Bandura's self-efficacy theory [18] and Knowles' principles [6], a unified SDL model tailored to clinical settings remains absent. The challenges in creating a universal SDL model stem from the dynamic nature of roles and responsibilities among healthcare professionals, which makes a "one-size-fits-all" solution impractical. In exploring SDL models beyond the traditional confines of healthcare, we observed a notable adaptation of these frameworks when applied within health professions education. While models originating from sectors like business, engineering, and technology emphasize efficiency, innovation, and scalability, adopting these models in healthcare education shifts focus towards patient-centered outcomes, ethical practice, and clinical safety. This adaptation reflects the unique priorities and ethical considerations inherent to healthcare, underscoring the need for SDL frameworks that foster knowledge acquisition and enhance clinical decision-making and ethical sensitivity. While universally applicable, the contextual, process, and personal dimensions of SDL models manifest distinctly within healthcare settings. For example, the contextual dimension in a corporate setting

might prioritize organizational culture and leadership styles, whereas in healthcare, it is more deeply intertwined with regulatory requirements, patient demographics, and interprofessional dynamics. This divergence highlights the malleable nature of SDL models, suggesting their potential for customization to meet the specific learning and operational needs of diverse professional fields.

Second, *on the practical level*, this study highlight actionable strategies to enhance SDL within healthcare settings. Our results stresses that institutions should prioritize SDL-friendly environments—such as providing protected learning time, access to digital tools, and mentorship programs. For example, a blended learning program during the COVID-19 pandemic demonstrated significant improvements in SDL competencies among nurses, highlighting the role of digital tools in informal learning [42]. On the other hand, mentorship programs pairing junior staff with experienced healthcare workers [52,56] seem to support goal-setting, access to resources, and constructive feedback. It may be hypothesized that hospitals encouraging open communication and offering digital learning platforms enable healthcare professionals to pursue SDL while balancing clinical responsibilities.

Digital tools play a crucial role in facilitating SDL by offering easy access to resources and enabling self-regulation. Examples include using mobile devices for informal learning [26] and integrating professional databases or online courses into daily practice [55]. However, challenges such as information overload and technical issues must be addressed to optimize their use. Moreover, peer collaboration during clinical rounds and team-based learning environments may significantly enhance SDL. Practical examples include clinical teams adapting to complex situations, such as the COVID-19 pandemic, where SDL was essential for learning under uncertainty [52]. In addition, SDL holds significant potential for application across diverse healthcare systems globally, offering a flexible and adaptable framework for continuous professional development. In high-income countries with well-resourced healthcare systems, SDL aligns demand to infrastructure that supports ongoing education, access to digital tools, and a culture of autonomy in learning [74,75]. Healthcare professionals in these settings often benefit from institutional support, advanced technology, and structured opportunities for SDL, making integrating SDL into their professional development pathways easier [76]. However, the applicability of SDL extends beyond these contexts, as it can be a critical educational strategy in low- and middle-income countries where resources for formal training may be limited. In such environments, SDL can be a cost-effective and scalable model, allowing healthcare professionals to engage in learning activities tailored to their needs, even without extensive institutional support or formal educational programs [77,78]. This adaptability highlights SDL's relevance in fostering lifelong learning and addressing the professional development needs of healthcare workers in diverse global contexts [76].

While universal SDL models are challenging to implement, frameworks tailored to specific educational levels (e.g., residents vs. practicing physicians) may enhance their applicability and impact. In countries with more hierarchical healthcare systems, implementing SDL may require a shift in organizational culture to promote greater autonomy and self-regulation among healthcare professionals [77]. Moreover, cultural norms regarding education and authority can influence how SDL is perceived and adopted. For example, in cultures that strongly emphasize deference to senior healthcare professionals, encouraging junior staff to take ownership of their learning may require specific interventions, such as mentoring programs or peer support networks [79]. Additionally, in resource-constrained settings, the availability of digital tools and reliable internet access may limit the effectiveness of SDL strategies that rely on online learning platforms. In these contexts, the success of SDL may depend on innovative approaches, such as using mobile learning or community-based learning networks [80]. Therefore, while SDL offers a universally applicable model for continuous professional development, its implementation must be carefully tailored to each healthcare system's socio-cultural and economic realities to ensure its effectiveness and sustainability.

Third, on the *descriptive level*, several findings are relevant: The diversity of study types and global participation—from North America, Europe, and Asia—underscores the universal relevance of SDL in advancing healthcare education and practice. These results highlight the necessity for applying SDL models in healthcare organizations to foster supportive environments that encourage continuous professional development and adaptability. The studies mainly focus on three

key aspects: the individual, the process, and the context. These aspects align with earlier research, showing consistent themes [6,81–83]. The importance of explicitly representing this common ground becomes evident through our literature analysis. This underscores the significance of documenting and addressing these fundamental components to ensure a comprehensive understanding and effective implementation of SDL across various contexts.

The analysis highlights that only about 1/3 of studies have explored how to implement SDL models in clinical settings. In addition, although these mentioned studies contain components that could fit into a model of SDL, a unified model for clinical settings was not identified.

This limited exploration of SDL implementation in clinical settings can be attributed to several factors. Conventional teaching methods prevalent in healthcare education, coupled with the demands of fast-paced clinical environments, might hinder the adoption of SDL [84]. Striking a balance between theoretical understanding and hands-on skills is challenging, while institutional policies and resource constraints often do not align with SDL models [85]. Educator readiness, assessment complexities, and the perception of learning outcomes and resistance to change further contribute to the limited exploration [86]. Lack of awareness, technology, and resources also play a role. Overcoming these challenges requires collaborative efforts to integrate SDL into clinical training effectively.

On the other hand, the diversity of individuals within medical education, including residents, practicing professionals, and doctors in training, poses a significant challenge in developing a "one-size-fits-all" SDL model. The dynamic nature of their roles, responsibilities, and learning needs makes creating a universally applicable model complex. While there have been indications that certain SDL models might be suitable for specific educational levels [8,70], such as practising physicians, the evidence supporting such adaptability needs to be firmly established.

Fourth, on the level of *professional perspective*, a notable gap exists in how healthcare organizations view SDL from a work psychological perspective. One potential explanation could be that healthcare organizations traditionally prioritize clinical skills and patient care over the psychological aspects of professional development. Additionally, the complex nature of healthcare environments, with high demands and limited resources, might lead organizations to focus more on immediate operational needs rather than long-term employee well-being and growth. Moreover, there might be limited awareness or understanding of how integrating work psychology principles into SDL strategies could improve performance, job satisfaction, and overall employee outcomes. Addressing this gap would require a shift in organizational mindset and a recognition of the profound impact of work psychological factors on promoting effective SDL and enhancing the healthcare workforce's capabilities and resilience. This could be due to SDL's newness in research, particularly in work and organizational psychology, compounded by a need for knowledge exchange between disciplines. The theory of self-directed learning processes (SDLPs) offers to drive SDL research in the HCP context [6,82,87].

This study also provides valuable insights related to RQ2 (*What are the key factors influencing the success of SDL in healthcare organizations, and how do these factors affect the experiences of healthcare professionals?)* Related to factors influencing SDL, we initially assumed that context was the most important. However, our comprehensive review shows that process-related and personal factors also significantly impact SDL. Unfortunately, a clear separation between these factors is often lacking, causing terminological overlap. Interestingly, a distinct categorization between factors that promote or hinder SDL is unclear, and was first suggested in this publication.

Personal factors, including motivation and prior experiences, significantly influence SDL. This can be attributed to the intricate interplay between individual attitudes, intrinsic drives, and accumulated learning history. Individuals who have successfully navigated SDL in the past may feel more confident and capable of managing their learning independently [88]. These experiences build a foundation of self-efficacy [89,90]. Personal factors, including motivation and prior experiences, significantly influence SDL. This can be attributed to the intricate interplay between individual attitudes, intrinsic drives, and accumulated learning history. Individuals who have successfully navigated SDL in the past may feel more confident and capable of managing their learning independently [88]. These experiences build a foundation of self-efficacy [89,90].

Contextual factors, studied extensively, reflect the interest in understanding how organizations can foster SDL in the workplace. Most significantly, peer support and digital tools' role in influencing SDL are significant in SDL endeavors. Peer support fosters shared experiences, collaborative learning, and emotional reinforcement [91,92], while digital tools facilitate convenient access to resources, aiding self-regulation and enhancing learning outcomes [11,93,94]. Understanding these factors is crucial for creating conducive environments that empower individuals in their SDL journeys. Therefore, the complex interplay of personal factors, along with the process and contextual influences, shapes the extent to which individuals engage in and benefit from SDL.

Social interactions and the learning environment also play pivotal roles in fostering SDL within healthcare settings. Peer collaboration during clinical rounds, for instance, allows healthcare professionals to exchange knowledge and discuss patient cases in real-time, fostering informal learning outside of formal training programs [75]. Similarly, mentorship programs, where junior staff are paired with experienced clinicians, provide structured support for setting learning goals, accessing resources, and receiving feedback, all of which encourage self-directed learning [95]. Additionally, institutions that offer digital learning platforms and online courses further promote SDL by allowing healthcare professionals to learn at their own pace, during work hours, or between clinical duties [55]. Hospitals that prioritize open communication, regular feedback, and protected learning time create environments conducive to SDL, making it easier for professionals to pursue their educational goals while maintaining their clinical responsibilities [56]. These practical examples demonstrate how both social interactions and supportive environments can significantly enhance the SDL process in healthcare settings.

Finally, while positive clinical outcomes such as enhanced quality of care and services could be anticipated, the current review reveals insufficient empirical data to substantiate these assumptions. Further empirical research is imperative to gauge the actual impact of SDL on employees. These insights are pivotal for the successful integration of SDL within healthcare organizations, underscoring the necessity for additional empirical data to refine existing approaches and validate assumptions.

## Limitations

Despite the highly contributive findings, several important limitations remain. First, although we aimed to explore the application and effectiveness of various SDL models within healthcare organizations, this was only partially achieved. Few studies have adopted a work organizational perspective, limiting our ability to fully analyze how organizational psychology impacts SDL strategies for HCPs. The lack of a clear definition for the SDL concept also posed challenges throughout this review, reflected in the low agreement between the raters during the initial article screening phase. This ambiguity led both reviewers to screen all articles to ensure consistency. Our comprehensive search strategy, which included related terms such as "self-regulated learning" and "informal learning," sometimes made it difficult to determine from the title and abstract whether SDL was being used as intended. We aimed to separate formal learning from informal SDL; however, the articles often lacked clarity. Second, relevant studies may have been missed despite our efforts, and some studies that used SDL models or frameworks may not have been explicitly identified. Nonetheless, we developed a search strategy with trained experts and librarians, refining it over several rounds to increase clarity. Although a relevant study may have been missed, we believe the results remain representative. Third, despite our intention to conduct a meta-analysis, the variability and quality of the included studies reversed this decision, as detailed in the Synthesis Results section. These challenges reflect the broader complexity of SDL in healthcare, where different professional contexts, educational environments, and learner characteristics shape the implementation and effectiveness of the models. While a meta-analysis could have provided stronger quantitative conclusions, by opting for a qualitative approach we were able to preserve each study's unique insights and offer a more nuanced interpretation of the findings. This allowed us to identify not only common themes and SDL facilitators, but also highlight gaps in the current research, particularly concerning the integration of SDL in different healthcare contexts. Moreover, the wide range of outcome measures and assessment tools used across studies limited our ability to conduct a meaningful

meta-analysis. Variations in study designs, SDL formats, and implementation strategies introduced inconsistencies that affected the comparability of the results, further complicating data synthesis. These factors may have affected the reliability of the findings, and potential bias must be considered when interpreting the results. Fourth, given the specific focus on SDL in HCPs' education, the findings may not be generalizable to other educational fields or professional settings. Finally, the studies included in this review were conducted in diverse geographical regions, which may limit the applicability of the findings to different healthcare systems or educational environments. Notwithstanding these limitations, to the best of our knowledge, this is the first systematic review to systematically explore how SDL is employed among HCPs and whether conceptual frameworks from work and organizational psychology have been integrated. We used several strategies to mitigate these limitations, including standardizing the SDL definition to enhance review consistency and integrating organizational psychology to broaden the analysis. Additional measures included refining the article screening protocol, conducting a comprehensive search, and extracting data to capture relevant studies and inferences using SDL models to ensure a comprehensive and accurate review. Effectively integrating SDL into clinical training requires collaborative efforts among stakeholders. Our study's deliberate inclusion of work and health psychology models, as outlined in the Methodology section, serves as a strategic criterion to bridge external knowledge and enhance the relevance of SDL within healthcare. Our findings underscore the applicability of diverse models and contribute to a more comprehensive discussion regarding overcoming the challenges of SDL implementation within healthcare organizations.

## Conclusion

The adaptation of SDL models from other sectors, such as business, engineering, and technology, into health profession education showcases a unique and strategic customization to meet specific needs within healthcare settings. These SDL models, which originally emphasized efficiency, innovation, and scalability at the individual level in their respective domains, take on new dimensions when applied to healthcare education. In the healthcare setting, the focus for SDL shifts towards outcomes directly relevant to the clinical setting, such as clinical safety, ethical practice, and patient-centered care. This reflects a broader and more holistic approach to health care education, underscoring the importance of outcomes beyond knowledge acquisition, including improving clinical decision-making and ethical sensitivity. A focus on ethics and safety is crucial in the healthcare setting, in which the consequences of decisions can significantly impact patient lives.

When SDL models are used in the corporate healthcare context, they must be integrated and aligned with regulatory requirements, patient demographics, and interprofessional dynamics. This involves understanding and navigating the complex landscape of healthcare regulations, catering to diverse patient needs, and fostering effective collaboration among HCPs.

In this regard, several strategies can be implemented to effectively support SDL in health profession education and practice.

• Providing tailored learning opportunities and timely feedback to help professionals understand their learning progress and the areas needing improvement.

• Encouraging a culture in which professionals can share insights and learn from each other, fostering a collaborative environment that enhances learning.

• Allowing professionals to engage in SDL activities during their work hours can help integrate learning directly into practice, making it more relevant and immediate.

• Leveraging technology, such as digital platforms and tools, can facilitate access to learning resources, streamline communication between learners and mentors, and support the tracking of learning progress.

**Supporting information**

**S1 Appendix. Study Protocol.**
(DOCX)

**S2 Appendix. Search Strategy.**
(DOCX)

**S3 Appendix. Theoretical Background.**
(DOCX)

**S4 Appendix. Extraction Form.**
(DOCX)

**S5 Appendix. List of Study Findings with Illustrations (Qualitative Results).**
(DOCX)

**S6 Appendix. Meta-aggregate Flowchart.**
(PDF)

**S1 Checklist. SDL_PRISMA_2020_checklist.**
(DOCX)

**S1 File. Table of excluded.**
(DOCX)

**S2 File. SX Appendix: Risk of Bias: ROBINS-1.**
(DOCX)

**Acknowledgments**

The authors thank Heidrun Ilonka Janka and Marc Simon von Gernler from the University Library of Bern for their help with the bibliographic search.

**Author contributions**

**Conceptualization:** Linda Krista, Felix Schmitz, Sissel Guttormsen.

**Data curation:** Joana Berger-Estilita, Artemisa Gogollari, Felix Schmitz, Sissel Guttormsen.

**Formal analysis:** Joana Berger-Estilita, Linda Krista, Artemisa Gogollari, Felix Schmitz, Achim Elfering, Sissel Guttormsen.

**Funding acquisition:** Sissel Guttormsen.

**Investigation:** Joana Berger-Estilita, Linda Krista, Artemisa Gogollari, Felix Schmitz, Achim Elfering, Sissel Guttormsen.

**Methodology:** Joana Berger-Estilita, Linda Krista, Artemisa Gogollari, Felix Schmitz, Achim Elfering, Sissel Guttormsen.

**Project administration:** Linda Krista, Felix Schmitz, Sissel Guttormsen.

**Resources:** Joana Berger-Estilita, Artemisa Gogollari.

**Software:** Artemisa Gogollari, Sissel Guttormsen.

**Supervision:** Achim Elfering, Sissel Guttormsen.

**Writing – original draft:** Joana Berger-Estilita, Linda Krista.

**Writing – review & editing:** Joana Berger-Estilita, Linda Krista, Artemisa Gogollari, Felix Schmitz, Achim Elfering, Sissel Guttormsen.

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
