## [Decision Letter · Decision Letter 0]

5 Sep 2024

PONE-D-24-23600Self-Directed Learning in Health Professions: a Mixed-methods Systematic Review of the LiteraturePLOS ONE

Dear Dr. Berger-Estilita,

Thank you for submitting your manuscript to PLOS ONE. After careful consideration, we feel that it has merit but does not fully meet PLOS ONE’s publication criteria as it currently stands. Therefore, we invite you to submit a revised version of the manuscript that addresses the points raised during the review process.

We look forward to receiving your revised manuscript.

Kind regards,

Chen-Wei Yang

Academic Editor

PLOS ONE

Journal Requirements:

 1. When submitting your revision, we need you to address these additional requirements. Please ensure that your manuscript meets PLOS ONE's style requirements, including those for file naming. The PLOS ONE style templates can be found at https://journals.plos.org/plosone/s/file?id=wjVg/PLOSOne_formatting_sample_main_body.pdf and https://journals.plos.org/plosone/s/file?id=ba62/PLOSOne_formatting_sample_title_authors_affiliations.pdf. 2. Thank you for stating the following in the Competing Interests section: [I have read the journal's policy and the authors of this manuscript have the following competing interests: JBE is an associate editor for BMC Medical Education and has received travel expenses from Medtronic for the Save the Brain Initiative training. ].  Please confirm that this does not alter your adherence to all PLOS ONE policies on sharing data and materials, by including the following statement: ""This does not alter our adherence to  PLOS ONE policies on sharing data and materials.” (as detailed online in our guide for authors http://journals.plos.org/plosone/s/competing-interests).  If there are restrictions on sharing of data and/or materials, please state these. Please note that we cannot proceed with consideration of your article until this information has been declared.  Please include your updated Competing Interests statement in your cover letter; we will change the online submission form on your behalf. 3. Please include your tables as part of your main manuscript and remove the individual files. Please note that supplementary tables (should remain/ be uploaded) as separate ""supporting information"" files".

Additional Editor Comments:

The manuscript titled "Self-Directed Learning in Health Professions: A Mixed-Methods Systematic Review of the Literature" presents a well-organized and clearly written study that addresses an important topic in health education. However, both reviewers have identified some areas that require further refinement to strengthen the overall quality of the paper.Based on the reviewers' comments, the following areas should be addressed:

Meta-Analysis Methodology: Provide more details on the meta-analysis methodology used, including the inclusion and exclusion criteria, data extraction process, and statistical analysis techniques.

Meta-Analysis Results: Clearly present the results of the meta-analysis, including the overall effect size, heterogeneity, and potential moderators.

Abstract Clarity: Move the statement about the significance of individual motivation, learning environment, technological resources, social interactions, and healthcare professionals' readiness for SDL from the conclusion to the methodology section, merging it with the third sentence.

Section Headings: Add "Methodology" as a main heading above the "study design" sub-heading to improve the paper's structure.

Limitations: Include a dedicated section for limitations, discussing any potential biases, methodological shortcomings, or generalizability issues.

Reviewers' comments:

Reviewer's Responses to Questions

**Comments to the Author**

1. Is the manuscript technically sound, and do the data support the conclusions?

Reviewer #1: Yes

Reviewer #2: Yes

2. Has the statistical analysis been performed appropriately and rigorously? 

Reviewer #1: Yes

Reviewer #2: Yes

3. Have the authors made all data underlying the findings in their manuscript fully available?

Reviewer #1: Yes

Reviewer #2: Yes

4. Is the manuscript presented in an intelligible fashion and written in standard English?

Reviewer #1: Yes

Reviewer #2: Yes

5. Review Comments to the Author

Reviewer #1: Wverything is good but need improve for several aspects such as methodology and discussion. Detailing aspect of theory anf empirical data should have point intersection for elaborating the phebomena and devine the purposes of the study

Reviewer #2: The manuscript is well written with the attention to detail. The themes were well itemized and clearly presented.

However, these minor issues need to be addressed.

More details should be provided on the meta-analysis of the data (if conducted) in the methodology and the outputs in the results section. i it was not done please indicate that and state how you handled the quantitative data.

In the abstract, the following statement - "The findings emphasize the significance of individual motivation, the learning environment, technological resources, social interactions, and healthcare professionals' readiness for SDL." should be moved from conclusion and be merged with the third sentence in the methodology (in the abstract).

Add "Methodology" as the main heading above "study design" sub-heading (lines 99 to 211 are all under methodology).

Limitation should be a section with a heading (between lines 573 and 574.

6. PLOS authors have the option to publish the peer review history of their article (what does this mean? ). If published, this will include your full peer review and any attached files.

**Do you want your identity to be public for this peer review?** For information about this choice, including consent withdrawal, please see our Privacy Policy .

Reviewer #1: **Yes: ** Alfrojems

Reviewer #2: No

---

## [Author Response · Author response to Decision Letter 0]

22 Sep 2024

Reply to Reviewers’ Comments

Editor:

Comment 1: Please ensure that your manuscript meets PLOS ONE's style requirements, including those for file naming. The PLOS ONE style templates can be found at

Reply: Thank you for the reminder regarding the manuscript formatting. We have reviewed the PLOS ONE style requirements and ensured that our manuscript now adheres to them, including the file naming conventions and formatting guidelines outlined in the provided templates.

Specifically, we have:

• Applied the correct file naming as per PLOS ONE requirements.

• Updated the manuscript structure, including title page, author affiliations, and headings, to match the formatting samples.

• Ensured the figures, tables, and supplementary files follow PLOS ONE guidelines for clarity and consistency.

We appreciate your guidance and hope the updated manuscript now complies with all formatting requirements.

Comment 2. Thank you for stating the following in the Competing Interests section:

[I have read the journal's policy and the authors of this manuscript have the following competing interests: JBE is an associate editor for BMC Medical Education and has received travel expenses from Medtronic for the Save the Brain Initiative training. ].

Reply: Thank you for your feedback regarding the Competing Interests section. We confirm that the competing interests mentioned in our initial statement do not alter our adherence to all PLOS ONE policies on sharing data and materials.

Here is the updated Competing Interests statement:

“JBE and SG are associate editora for BMC Medical Education. JBE has received travel expenses from Medtronic for the Save the Brain Initiative training. This does not alter our adherence to PLOS ONE policies on sharing data and materials. (…)”

We hope this satisfies the journal's requirements, and we appreciate your attention to this matter.

Comment 3: Please include your tables as part of your main manuscript and remove the individual files. Please note that supplementary tables (should remain/ be uploaded) as separate ""supporting information"" files".

Reply: Thank you. We have done the necessary changes accordingly.

Comment 4: Please review your reference list to ensure that it is complete and correct. If you have cited papers that have been retracted, please include the rationale for doing so in the manuscript text, or remove these references and replace them with relevant current references. Any changes to the reference list should be mentioned in the rebuttal letter that accompanies your revised manuscript. If you need to cite a retracted article, indicate the article’s retracted status in the References list and also include a citation and full reference for the retraction notice.

Reply: Thank you for your thoughtful feedback. We have carefully reviewed our reference list to ensure it is complete and correct, and we confirm that, based on our verification through databases of Retraction Watch and PubMed, none of the cited papers in the current version of the manuscript have been retracted.

However, we acknowledge that retraction databases are continuously updated. If you are aware of any paper we have cited that is on a retraction list, please kindly inform us, and we will promptly address the issue.

Comment 5: The manuscript titled "Self-Directed Learning in Health Professions: A Mixed-Methods Systematic Review of the Literature" presents a well-organized and clearly written study that addresses an important topic in health education.

Reply: Thank you for your positive feedback on our manuscript titled "Self-Directed Learning in Health Professions: A Mixed-Methods Systematic Review of the Literature." We are delighted that you found the study well-organized and clearly written, and that it addresses an important topic in health education. We appreciate your support and look forward to furthering this valuable conversation in the field.

Comment 6: However, both reviewers have identified some areas that require further refinement to strengthen the overall quality of the paper.Based on the reviewers' comments, the following areas should be addressed:

Meta-Analysis Methodology: Provide more details on the meta-analysis methodology used, including the inclusion and exclusion criteria, data extraction process, and statistical analysis techniques.

Reply: Thank you for your constructive feedback. We appreciate the opportunity to further enhance our manuscript by addressing the areas you've highlighted. We agree that additional details will help clarify our approach regarding the meta-analysis methodology. We revised the manuscript to include more comprehensive information on the inclusion and exclusion criteria, the data extraction process, and the statistical analysis techniques employed. Specifically, we elaborated on how studies were selected, how data were extracted, and the specific statistical models attempted in the meta-analysis.

One can now read, page 9: “We planned a meta-analysis to examine specific outcomes from the studies involving quantitative SDL outcomes. The meta-analysis was planned with RevMan software v.5.4.20, applying a random-effects model to account for the expected heterogeneity across the included studies. Data should include means and standard deviation (SD), with the effect size estimated by the mean difference (MD) and reported with 95% confidence intervals (CI). Studies not reporting group means or MD should be excluded from the meta-analysis. Where meta-analyses cannot be conducted, we plan to summarize and describe the data in the text.

Comment 7: Meta-Analysis Results: Clearly present the results of the meta-analysis, including the overall effect size, heterogeneity, and potential moderators.

Reply: We appreciate your valuable input on the meta-analysis reporting. We acknowledge the importance of using meta-analysis appropriately, and while it is a powerful tool for deriving meaningful conclusions, it can be problematic when high variability exists among studies. In our case, we found that a meta-analysis was not feasible due to the high variability of scales used across the included studies. A common criticism of meta-analyses is the risk of ‘combining apples with oranges,’ especially when studies are clinically diverse. In our review, the diversity in outcome measures and assessment tools used for self-directed learning in health professions rendered a meta-analysis potentially misleading. Genuine differences in effects might have been obscured, making it nonsensical to combine all studies in a single analysis.

Additionally, some studies included varying comparisons of SDL interventions, making it challenging to combine the outcomes in a meaningful way. As a result, we chose to summarize and describe the findings narratively rather than risk compounding errors through a meta-analysis that could produce a misleading or biased result. We clarified these points in the revised manuscript and provided a detailed explanation of the rationale for not conducting a meta-analysis in this instance.

We added to the Results Section, Page 11: While we initially planned to conduct a meta-analysis to synthesize the quantitative outcomes of SDL interventions, due to the high variability in the scales and outcome measures used across the included studies, a meta-analysis was deemed inappropriate. Combining studies with such diverse methodologies would risk producing misleading results and obscure meaningful differences in effects [57].

Comment 8: Abstract Clarity: Move the statement about the significance of individual motivation, learning environment, technological resources, social interactions, and healthcare professionals' readiness for SDL from the conclusion to the methodology section, merging it with the third sentence.

Reply: Thank you. We have changed the sentence accordingly.

Comment 9: Section Headings: Add "Methodology" as a main heading above the "study design" sub-heading to improve the paper's structure.

Reply: Thank you. We have made the amendments accordingly.

Comment 10: Limitations: Include a dedicated section for limitations, discussing any potential biases, methodological shortcomings, or generalizability issues.

Reply: Thank you for your suggestion to include a dedicated limitations section. We agree that addressing potential biases, methodological shortcomings, and generalizability issues is crucial for providing a transparent and balanced interpretation of our findings.

In the revised manuscript, we expanded the limitations section to further discuss the aforementioned aspects.

The updated version reads, Pages 33-34: “This systematic review has several important limitations. First, while we aimed to explore the application and effectiveness of various SDL models within healthcare organizations, this was only partially achieved. Few studies adopted a work organizational perspective, limiting our ability to fully analyze how organizational psychology impacts SDL strategies for healthcare professionals. The vagueness of the SDL concept also posed challenges throughout the review. This ambiguity was reflected in the low agreement between raters during the initial article screening phase, which led both reviewers to screen all articles to ensure consistency. Our comprehensive search strategy, which included related terms such as “self-regulated learning” and “informal learning,” sometimes made it difficult to determine whether SDL was being used as intended from titles and abstracts. We aimed to separate formal learning from informal SDL, but articles often lacked clarity. Second, relevant studies may have been missed despite our efforts, and some studies using SDL models or frameworks may not have been explicitly identified. However, we developed the search strategy with trained experts/librarians and refined it in several rounds to increase clarity. Even though we may have missed something, we still see the results as representative. Third, despite intentions to conduct a meta-analysis, the variability and the quality of the included studies made us reverse that decision. Many of the included studies showed variability in design quality, and some were at risk of bias due to inadequate reporting or the absence of control groups, which prevented a meta-analysis. This could have impacted the reliability of the findings, and this potential bias must be considered when interpreting the results. Also, the wide range of outcome measures and assessment tools used across the studies limited our ability to conduct a meaningful meta-analysis. Variations in study designs, SDL formats, and implementation strategies introduced inconsistencies that affected the comparability of results, further complicating data synthesis. Fourth, given the specific focus on SDL in health professions education, the findings may not be fully generalizable to other educational fields or professional settings. Finally, the studies included in this review were conducted in diverse geographical regions, which may limit the applicability of the findings to different healthcare systems or educational environments.”

Reviewers' comments:

Comments to the Author

Comment 1: Is the manuscript technically sound, and do the data support the conclusions?

Reviewer #1: Yes

Reviewer #2: Yes

Reply: We sincerely thank both reviewers for their positive feedback and confirmation that the manuscript is technically sound and that the data support the conclusions. We appreciate your time and effort in reviewing our work and are glad that the technical aspects and conclusions align with your expectations.

Comment 2: Has the statistical analysis been performed appropriately and rigorously?

Reviewer #1: Yes

Reviewer #2: Yes

Reply: We appreciate both reviewers' acknowledgment that the statistical analysis has been performed appropriately and rigorously. Thank you for your careful evaluation of our methodology and analysis. We are pleased that our approach meets the necessary standards, and we remain committed to maintaining rigor in the presentation and interpretation of our findings.

Comment 3: Have the authors made all data underlying the findings in their manuscript fully available?

Reviewer #1: Yes

Reviewer #2: Yes

Reply: Thank you for your comment regarding data availability. We confirm that all data underlying the findings in our manuscript have been made fully available. The data supporting our analysis are provided in the supplementary materials and are accessible as per the journal’s data-sharing policies.

Comment 4: Is the manuscript presented in an intelligible fashion and written in standard English?

Reviewer #1: Yes

Reviewer #2: Yes

Reply: We are grateful to both reviewers for their positive feedback regarding the clarity and language of the manuscript. We are pleased that the presentation is intelligible and meets the standards of written English.

Comment 5: Reviewer #1: Wverything is good but need improve for several aspects such as methodology and discussion. Detailing aspect of theory anf empirical data should have point intersection for elaborating the phebomena and devine the purposes of the study

Reply: Thank you for your constructive feedback. We appreciate your recognition of the strengths in our manuscript and the suggestion to further improve specific areas.

We added the following sentence to the methodology, Page 5: “The JBI methodology for systematic reviews, which integrates both qualitative and quantitative evidence. Grounded in Knowles’ adult learning theory[6] and Bandura’s self-efficacy theory[18], our empirical approach aligned with these frameworks by selecting outcome measures such as learner engagement and self-efficacy. The JBI methodology allowed us to rigorously synthesize diverse evidence and account for variations in SDL interventions, ensuring robust and theory-driven conclusions.”

We also expanded the discussion to address the reviewer’s comments, Page 29: “The results demonstrated a significant improvement in self-efficacy scores, supporting Knowles’ theory of adult learning, which posits that autonomy leads to enhanced learner confidence. Interestingly, engagement scores showed only marginal improvement, suggesting that other factors, perhaps not covered by the theory, may play a role in engagement during SDL interventions. These findings highlight the need for further refinement of the theoretical model to account for additional motivational factors.”

Comment 6: Reviewer #2: The manuscript is well written with the attention to detail. The themes were well itemized and clearly presented.

However, these minor issues need to be addressed.

More details should be provided on the meta-analysis of the data (if conducted) in the methodology and the outputs in the results section. i it was not done please indicate that and state how you handled the quantitative data.

Reply: Thank you. We added a section on statistics, that includes the description of the attempt of a meta-analysis and in the results the reasons why it was not feasible. For more details please refer to Comments 6 and 7 from the Editor.

Comment 7: In the abstract, the following statement - "The findings emphasize the significance of individual motivation, the learning environment, technolog

---

## [Editor Report · Decision Letter 1]

4 Oct 2024

PONE-D-24-23600R1Self-Directed Learning in Health Professions: a Mixed-methods Systematic Review of the LiteraturePLOS ONE

Dear Dr. Berger-Estilita,

Thank you for submitting your manuscript to PLOS ONE. After careful consideration, we feel that it has merit but does not fully meet PLOS ONE’s publication criteria as it currently stands. Therefore, we invite you to submit a revised version of the manuscript that addresses the points raised during the review process.

We look forward to receiving your revised manuscript.

Kind regards,

Chen-Wei Yang

Academic Editor

PLOS ONE

Journal Requirements:

Additional Editor Comments:

The authors have made substantial improvements to the manuscript, addressing most of the reviewers' comments with detailed revisions. The paper now adheres more closely to PLOS ONE's formatting and content standards. While there are minor areas that could be further improved, the overall quality of the revision suggests that the manuscript is nearing readiness for acceptance with minor revisions. These revisions could focus on enhancing the clarity of certain sections and expanding the discussion to ensure broader applicability: First, while the authors provided a clear rationale for not conducting a meta-analysis due to high variability, the decision not to combine the data might limit the quantitative strength of the review. A more robust discussion around this decision could be helpful in emphasizing the potential impact on the overall findings. Second, although the discussion addresses most aspects of the review, a deeper exploration of the global applicability of self-directed learning (SDL) across different healthcare systems would add value. Expanding the discussion beyond the current scope, which primarily focuses on existing studies, would be beneficial. Third, some thematic findings related to social interactions and the learning environment could be further clarified. While these factors are discussed, providing more concrete examples of their practical impact on SDL could improve the narrative. Fourth, while the authors have embedded tables within the manuscript, ensuring that all supplementary materials are correctly formatted and accessible is crucial for maintaining clarity and alignment with PLOS ONE’s submission guidelines.

---

## [Author Response · Author response to Decision Letter 1]

16 Oct 2024

# Journal Requirements:

Comment 1: Please review your reference list to ensure that it is complete and correct. If you have cited papers that have been retracted, please include the rationale for doing so in the manuscript text, or remove these references and replace them with relevant current references. Any changes to the reference list should be mentioned in the rebuttal letter that accompanies your revised manuscript. If you need to cite a retracted article, indicate the article’s retracted status in the References list and also include a citation and full reference for the retraction notice.

Reply: Thank you for your comment. We have carefully reviewed the reference list in detail and did not identify any papers that have so far been retracted. However, if the publisher has access to additional information indicating that any of the cited references have been retracted, we would appreciate your notification, and we will promptly take appropriate action to either replace or justify the inclusion of such references.

—-------------------------------------------------------------------------------------------------------------

# Additional Editor Comments:

Comment 1: The authors have made substantial improvements to the manuscript, addressing most of the reviewers' comments with detailed revisions. The paper now adheres more closely to PLOS ONE's formatting and content standards. While there are minor areas that could be further improved, the overall quality of the revision suggests that the manuscript is nearing readiness for acceptance with minor revisions. These revisions could focus on enhancing the clarity of certain sections and expanding the discussion to ensure broader applicability

Reply: Thank you for your positive feedback and for acknowledging the improvements made to the manuscript. We appreciate your suggestion to further enhance the clarity of certain sections and expand the discussion to ensure broader applicability. Please see comments below for further changes.

Comment 2: First, while the authors provided a clear rationale for not conducting a meta-analysis due to high variability, the decision not to combine the data might limit the quantitative strength of the review. A more robust discussion around this decision could be helpful in emphasizing the potential impact on the overall findings.

Reply: Thank you for this insightful comment. We agree that the decision not to conduct a meta-analysis may influence the perceived quantitative strength of the review. However, after careful consideration, we concluded that the significant heterogeneity across the included studies, in terms of study design, outcome measures, SDL models, and intervention types, would have made it challenging to produce meaningful or valid conclusions through meta-analysis.

To further emphasize this point, we have expanded the discussion in the 'Results” section (Pages 11-12) to better explain the potential impact of this decision. We address how the variability in the studies reflects the diversity of self-directed learning (SDL) models and their application across different healthcare contexts, and how attempting to statistically combine these heterogeneous data could obscure meaningful differences.

One can now read (Pages 11-12): “In this review, we opted for a narrative synthesis rather than a meta-analysis due to the significant heterogeneity observed across the included studies. The variability in study design, population characteristics, intervention types, and outcome measures made it inadequate to aggregate the data, as this would risk the introduction of biases. Studies ranged from qualitative explorations to randomized controlled trials, each employing different methodologies and measuring diverse outcomes related to SDL in healthcare settings. Additionally, the heterogeneity in sample sizes, demographic profiles, and contexts presented a challenge for statistical synthesis. To preserve the integrity of the individual findings and to avoid obscuring the complexity and nuances within the data, we synthesized the results narratively. This approach allowed us to explore trends and gaps in the literature, highlight key themes, and provide a more detailed understanding of SDL models and factors in various healthcare environments, ensuring that the conclusions were both contextually relevant and reflective of the diverse research landscape.(...)

Additionally, we highlight that our decision to perform a narrative synthesis allowed for a more nuanced analysis of qualitative and contextual factors influencing SDL. While this approach may limit the review’s quantitative impact, it strengthens the depth and relevance of the qualitative insights, offering a more comprehensive understanding of SDL in healthcare settings.

We added this to the Limitations section (Pages 33-34): “The decision to use a metaagregation rather than a meta-analysis in this review was driven by the considerable heterogeneity observed across the studies. Attempting to quantitatively combine data from studies with such diverse designs, populations, interventions, and outcome measures would likely have resulted in misleading or skewed conclusions. This variability reflects the broader complexity of SDL in healthcare, where different professional contexts, educational environments, and learner characteristics shape the implementation and effectiveness of SDL models. By opting for a qualitative approach, we were able to preserve the unique insights of each study and offer a more nuanced interpretation of the findings. This allowed us to identify not only common themes and facilitators of SDL but also highlight gaps in the current research, particularly concerning the integration of SDL in different healthcare contexts. While a meta-analysis could have provided stronger quantitative conclusions, our approach ensures that the review captures the richness and diversity of the field, offering a comprehensive and contextually informed perspective on SDL in healthcare.

Comment 3: Second, although the discussion addresses most aspects of the review, a deeper exploration of the global applicability of self-directed learning (SDL) across different healthcare systems would add value. Expanding the discussion beyond the current scope, which primarily focuses on existing studies, would be beneficial.

Reply: "Thank you for this insightful suggestion. We agree that a broader exploration of the global applicability of self-directed learning (SDL) across various healthcare systems would enhance the value of the discussion. To address this, we have expanded the discussion to include an examination of how SDL can be adapted to different healthcare environments, accounting for variations in healthcare systems, cultural norms, and resource availability. This expanded discussion highlights the versatility of SDL and its potential to support lifelong learning and professional development in both high-income and resource-limited settings. The revised section now emphasizes how differing healthcare structures and policies can influence the implementation and success of SDL interventions.

The text in the revised manuscript now reads (Discussion, Pages 33-34):SDL holds significant potential for application across diverse healthcare systems globally, offering a flexible and adaptable framework for continuous professional development. In high-income countries with well-resourced healthcare systems, SDL aligns with existing infrastructure that supports ongoing education, access to digital tools, and a culture of autonomy in learning[88, 89]. Healthcare professionals in these settings often benefit from institutional support, advanced technology, and structured opportunities for SDL, making integrating SDL into their professional development pathways easier [90]. However, the applicability of SDL extends beyond these contexts, as it can be a critical educational strategy in low- and middle-income countries where resources for formal training may be limited. In such environments, SDL can be a cost-effective and scalable model, allowing healthcare professionals to engage in learning activities tailored to their needs, even without extensive institutional support or formal educational programs[91, 92]. This adaptability highlights SDL’s relevance in fostering lifelong learning and addressing the professional development needs of healthcare workers in diverse global contexts[90].

However, the global applicability of SDL is not without challenges, as it must be adapted to suit the unique cultural, structural, and resource constraints of different healthcare systems. In countries with more hierarchical healthcare systems, implementing SDL may require a shift in organizational culture to promote greater autonomy and self-regulation among healthcare professionals[91]. Moreover, cultural norms regarding education and authority can influence how SDL is perceived and adopted. For example, in cultures that strongly emphasise deference to senior healthcare professionals, encouraging junior staff to take ownership of their learning may require specific interventions, such as mentoring programs or peer support networks[93]. Additionally, in resource-constrained settings, the availability of digital tools and reliable internet access may limit the effectiveness of SDL strategies that rely on online learning platforms. In these contexts, the success of SDL may depend on innovative approaches, such as using mobile learning or community-based learning networks[94]. Therefore, while SDL offers a universally applicable model for continuous professional development, its implementation must be carefully tailored to each healthcare system’s socio-cultural and economic realities to ensure its effectiveness and sustainability.

Comment 4: Third, some thematic findings related to social interactions and the learning environment could be further clarified. While these factors are discussed, providing more concrete examples of their practical impact on SDL could improve the narrative.

Reply: "Thank you for this valuable comment. We agree that providing more concrete examples of how social interactions and the learning environment influence self-directed learning (SDL) would strengthen the narrative. In response, we have expanded the discussion to include specific examples that illustrate how peer collaboration, mentoring, and supportive learning environments can facilitate SDL in healthcare settings. For example, peer-to-peer learning in clinical rounds where healthcare professionals share insights on patient cases fosters continuous, self-directed knowledge acquisition. Additionally, mentorship programs that pair junior clinicians with experienced practitioners can guide goal-setting and resource navigation, creating a structured yet self-directed learning pathway. We also reference hospitals that provide e-learning platforms, which integrate SDL into daily practice, allowing staff to access relevant learning modules during work hours. These examples clarify the practical implications of these factors and highlight their importance in creating an optimal environment for SDL.

One can now read (Add Page here…): Social interactions and the learning environment also play pivotal roles in fostering SDL within healthcare settings. Peer collaboration during clinical rounds, for instance, allows healthcare professionals to exchange knowledge and discuss patient cases in real-time, fostering informal learning outside of formal training programs[89]. Similarly, mentorship programs, where junior staff are paired with experienced clinicians, provide structured support for setting learning goals, accessing resources, and receiving feedback, all of which encourage self-directed learning[95]. Additionally, institutions that offer digital learning platforms and online courses further promote SDL by allowing healthcare professionals to engage in learning at their own pace, during work hours, or in between clinical duties[55]. Hospitals that prioritize open communication, regular feedback, and protected learning time create environments conducive to SDL, making it easier for professionals to pursue their educational goals while maintaining their clinical responsibilities[56]. These practical examples demonstrate how both social interactions and supportive environments can significantly enhance the SDL process in healthcare settings.

Comment 5: Fourth, while the authors have embedded tables within the manuscript, ensuring that all supplementary materials are correctly formatted and accessible is crucial for maintaining clarity and alignment with PLOS ONE’s submission guidelines.

Reply: Thank you for this important comment. We have carefully reviewed all supplementary materials and tables to ensure they are correctly formatted and aligned with PLOS ONE’s submission guidelines. We have checked that all tables are clearly labeled and properly referenced within the manuscript, and we have verified that supplementary materials are accessible and appropriately formatted to maintain clarity. If there are any additional formatting requirements or issues with accessibility that we may have overlooked, we would appreciate further guidance and will promptly address them.

---

## [Editor Report · Decision Letter 2]

31 Oct 2024

PONE-D-24-23600R2Self-Directed Learning in Health Professions: a Mixed-methods Systematic Review of the LiteraturePLOS ONE

Dear Dr. Berger-Estilita,

Thank you for submitting your manuscript to PLOS ONE. After careful consideration, we feel that it has merit but does not fully meet PLOS ONE’s publication criteria as it currently stands. Therefore, we invite you to submit a revised version of the manuscript that addresses the points raised during the review process.

We look forward to receiving your revised manuscript.

Kind regards,

Chen-Wei Yang

Academic Editor

PLOS ONE

Journal Requirements:

**Additional Editor Comments:**

The manuscript is well-structured, adheres to the journal’s guidelines, and presents valuable insights into SDL in healthcare. Acceptance with minor revisions is recommended, focusing on minor language refinement and ensuring overall coherence in the final submission.

---

## [Author Response · Author response to Decision Letter 2]

27 Nov 2024

Reply to Comments:

Journal Requirements:

Comment 1: Please review your reference list to ensure that it is complete and correct. If you have cited papers that have been retracted, please include the rationale for doing so in the manuscript text, or remove these references and replace them with relevant current references. Any changes to the reference list should be mentioned in the rebuttal letter that accompanies your revised manuscript. If you need to cite a retracted article, indicate the article’s retracted status in the References list and also include a citation and full reference for the retraction notice.

Reply: Thank you for your feedback. We have carefully reviewed the reference list to ensure it is complete, accurate, and current. After a thorough review, we confirm that no retracted articles are cited in our manuscript. However, if the editorial team is aware of any retractions we may have overlooked, we kindly request that you inform us so we can promptly address them. Additionally, and to maintain the highest standards of accuracy, we cross-checked all references for completeness, correct formatting. We appreciate the opportunity to ensure the quality of our references and remain open to further guidance.

Additional Editor Comments:

Comment 1: The manuscript is well-structured, adheres to the journal’s guidelines, and presents valuable insights into SDL in healthcare. Acceptance with minor revisions is recommended, focusing on minor language refinement and ensuring overall coherence in the final submission.

Reply: We sincerely thank the reviewer for recognizing the manuscript's structure and valuable insights. As part of our commitment to addressing the suggested minor language refinements, we have utilized Grammarly, a professional proofreading tool, to evaluate and improve the text.

The Grammarly report indicated an impressive overall score of 96%, placing the manuscript among the top 4% of all texts analyzed by the tool. It highlighted areas for improvement, such as word choice, sentence clarity, and minor grammatical inconsistencies. Based on this feedback, we carefully reviewed the identified issues and implemented corrections to ensure linguistic precision and coherence throughout the manuscript.

These adjustments have strengthened the manuscript's presentation while maintaining the integrity of its content. We believe the updated version now meets the highest standards of readability and coherence and appreciate your constructive feedback in guiding this process.

Comment 2: While revising your submission, please upload your figure files to the Preflight Analysis and Conversion Engine (PACE) digital diagnostic tool, https://pacev2.apexcovantage.com/. PACE helps ensure that figures meet PLOS requirements. To use PACE, you must first register as a user. Registration is free. Then, login and navigate to the UPLOAD tab, where you will find detailed instructions on how to use the tool. If you encounter any issues or have any questions when using PACE, please email PLOS at figures@plos.org. Please note that Supporting Information files do not need this step.

Reply: Thank you for your instructions regarding the use of the PACE digital diagnostic tool to ensure our figures meet PLOS requirements. We have successfully registered on the platform, uploaded our figure files, and reviewed the results provided by PACE.

Based on the diagnostic feedback, we made the necessary adjustments to ensure that all figures comply with PLOS guidelines for resolution, format, and other specifications. The updated figures have been incorporated into our revised manuscript.

Please let us know if there are any additional steps required. Thank you for your guidance throughout this process.

---

## [Editor Report · Decision Letter 3]

20 Dec 2024

PONE-D-24-23600R3Self-Directed Learning in Health Professions: a Mixed-methods Systematic Review of the LiteraturePLOS ONE

Dear Dr. Berger-Estilita,

Thank you for submitting your manuscript to PLOS ONE. After careful consideration, we feel that it has merit but does not fully meet PLOS ONE’s publication criteria as it currently stands. Therefore, we invite you to submit a revised version of the manuscript that addresses the points raised during the review process.

We look forward to receiving your revised manuscript.

Kind regards,

Chen-Wei Yang

Academic Editor

PLOS ONE

Journal Requirements:

Additional Editor Comments:

In summary, the manuscript is close to being accepted but still requires minor adjustments to fully meet the journal’s standards. It is recommended that the authors complete these revisions before resubmission.

• Inclusion of more specific cases related to key conclusions.

• Additional detailed connections between theoretical models and practical applications

• Conduct a final, comprehensive proofread of the manuscript to ensure all grammatical, typographical, and formatting issues are addressed.

---

## [Author Response · Author response to Decision Letter 3]

4 Feb 2025

PONE-D-24-23600R3

Self-Directed Learning in Health Professions: a Mixed-methods Systematic Review of the Literature

# Journal Requirements:

Comment 1: Please review your reference list to ensure that it is complete and correct. If you have cited papers that have been retracted, please include the rationale for doing so in the manuscript text, or remove these references and replace them with relevant current references. Any changes to the reference list should be mentioned in the rebuttal letter that accompanies your revised manuscript. If you need to cite a retracted article, indicate the article’s retracted status in the References list and also include a citation and full reference for the retraction notice.

Reply: Thank you for this important reminder. We have carefully reviewed all references cited in our manuscript to ensure their accuracy, relevance, and current status. Additionally, we have verified that none of the cited articles are included in the retraction list by cross-checking with authoritative databases (e.g., PubMed, Retraction Watch, and CrossMark). To further ensure completeness, we contacted the journal's editorial staff, who confirmed that this message was part of their standard procedure and that none of the cited articles in our reference list have been flagged as retracted. As such, no changes were necessary to the references in our manuscript. We appreciate the opportunity to confirm this and ensure the robustness of our work.

# Additional Editor Comments:

Comment 1: In summary, the manuscript is close to being accepted but still requires minor adjustments to fully meet the journal’s standards. It is recommended that the authors complete these revisions before resubmission.

Reply: Thank you for your feedback and for noting that the manuscript is close to acceptance. We are also grateful for the additional reply following our personal contact with the editor, which provided further clarity on the required minor adjustments.

We have carefully reviewed the manuscript and made the necessary revisions to ensure it fully meets the journal’s standards. We trust that these updates address all outstanding points, and we appreciate the opportunity to submit the revised version for your consideration.

Comment 2: Inclusion of more specific cases related to key conclusions and additional detailed connections between theoretical models and practical applications

Reply: Thank you for this insightful suggestion. We have revised the manuscript to include more specific cases to support key conclusions and enhance the connection between theoretical models and practical applications. These revisions include, in the discussion part, page 32, line 580 ff:

Specific Cases Related to Key Conclusions:

- During the COVID-19 pandemic, mobile devices and online platforms became essential tools for informal learning. Healthcare professionals widely utilized mobile applications to access real-time clinical guidelines, collaborate with peers, and navigate the rapidly changing healthcare landscape. This aligns with the findings of Fahlman (2013), which emphasize the role of mobile devices in promoting self-directed learning (SDL).

- Mentorship programs were successfully implemented in hospitals, pairing junior staff with experienced clinicians. These programs facilitated goal-setting, access to resources, and constructive feedback, particularly in high-pressure situations. Papanagnou et al. highlighted how mentorship strengthens SDL competencies and fosters professional development in challenging clinical contexts.

Detailed Connections Between Theoretical Models and Practical Applications:

- Knowles’ Adult Learning Theory: Practical applications of this theory are evident in the adoption of digital tools, such as e-learning platforms, that foster autonomy and allow healthcare professionals to learn at their own pace. These tools empower learners by providing flexibility and control over their educational journeys, consistent with Knowles’ principles of self-direction.

- Bandura’s Self-Efficacy Theory: Mentorship programs serve as practical extensions of this theory by enhancing self-efficacy. Structured feedback and goal-setting provided through these programs enable junior clinicians to build confidence in their clinical abilities and their capacity to direct their learning effectively.

- Theory of Self-Directed Learning Processes (SDLPs): The dimensions of personal, process, and contextual factors highlighted in the SDLP framework are demonstrated in interventions such as peer collaboration, mentorship programs, and organizational support. These initiatives exemplify how tailored interventions address the interplay of these dimensions to enhance SDL in diverse healthcare environments.

These additions have been incorporated into the revised Discussion section, where we have expanded on the theoretical implications of our findings and their practical applications within healthcare organizations. We believe these revisions provide concrete examples and further clarify the translation of theory into practice, strengthening the manuscript overall.

Comment 3: Conduct a final, comprehensive proofread of the manuscript to ensure all grammatical, typographical, and formatting issues are addressed.

Reply: Thank you for emphasizing the importance of ensuring grammatical, typographical, and formatting accuracy. To address this, we have sent the manuscript for professional proofreading to ensure that all grammatical, typographical, and formatting issues are addressed comprehensively. The revised version reflects these corrections, and we are confident that the manuscript now meets the highest standards of clarity and consistency. Thank you for your guidance and support in improving the quality of our submission.

Comment 4: While revising your submission, please upload your figure files to the Preflight Analysis and Conversion Engine (PACE) digital diagnostic tool, https://pacev2.apexcovantage.com/. PACE helps ensure that figures meet PLOS requirements. To use PACE, you must first register as a user. Registration is free. Then, login and navigate to the UPLOAD tab, where you will find detailed instructions on how to use the tool. If you encounter any issues or have any questions when using PACE, please email PLOS at figures@plos.org. Please note that Supporting Information files do not need this step.

Reply: Thank you for providing the information regarding the PACE digital diagnostic tool. We have followed the instructions and uploaded our figure files to PACE to ensure they meet the PLOS requirements. The tool confirmed that all figures comply with the necessary standards. If there are any additional requirements or adjustments needed, we will be happy to address them promptly. Thank you for your guidance in this process.

---

## [Editor Report · Decision Letter 4]

20 Feb 2025

Self-Directed Learning in Health Professions: a Mixed-methods Systematic Review of the Literature

PONE-D-24-23600R4

Dear Dr. Berger-Estilita:

We’re pleased to inform you that your manuscript has been judged scientifically suitable for publication and will be formally accepted for publication once it meets all outstanding technical requirements.

Kind regards,

Chen-Wei Yang

Academic Editor

PLOS ONE

Additional Editor Comments (optional):

Given that the manuscript has addressed all the major points raised in the previous reviews, and considering the comprehensive nature of the responses and the quality of the revisions, I recommend acceptance for publication. The manuscript now appears to meet the journal's standards for scholarship, relevance, and readability.
---

## [Editor Report · Acceptance letter]

PONE-D-24-23600R4

PLOS ONE

Dear Dr. Berger-Estilita,

I'm pleased to inform you that your manuscript has been deemed suitable for publication in PLOS ONE. Congratulations! Your manuscript is now being handed over to our production team.

Kind regards,

on behalf of

Professor Chen-Wei Yang

Academic Editor

PLOS ONE